# Fatigue Life Assessment of Refill Friction Stir Spot Welded Alclad 7075-T6 Aluminium Alloy Joints

**Andrzej Kubit** [1,*] **, Mateusz Drabczyk** [2] **, Tomasz Trzepiecinski** [3] **, Wojciech Bochnowski** [4] **,**
**Ľuboš Kaščák** [5] **and Jan Slota** [5]

[1]   Department of Manufacturing and Production Engineering, Rzeszow University of Technology, al. Powst. Warszawy 8, 35-959 Rzeszów, Poland

[2]   Institute of Technology, Faculty of Mathematics and Natural Sciences, University of Rzeszów, al. Rejtana 16c, 35-959 Rzeszow, Poland; mat.drabczyk@gmail.com

[3]   Department of Materials Forming and Processing, Rzeszow University of Technology, al. Powst. Warszawy 8, 35-959 Rzeszów, Poland; tomtrz@prz.edu.pl

[4]   Centre for Innovative Technologies, University of Rzeszów, ul. Pigonia 1, 35-959 Rzeszów, Poland; wobochno@ur.edu.pl

[5]   Institute of Technology and Material Engineering, Faculty of Mechanical Engineering, Technical University of Košice, Mäsiarska 74, 040 01 Košice, Slovakia; lubos.kascak@tuke.sk (Ľ.K.); jan.slota@tuke.sk (J.S.)

\*   Correspondence: akubit@prz.edu.pl; Tel.: +48-17-743-2019

**Abstract:** Refill Friction Stir Spot Welding (RFSSW) shows great potential to be a replacement for single-lap joining techniques such as riveting or resistance spot welding used in the aircraft industry. In this paper, the fatigue behaviour of RFSSW single-lap joints is analysed experimentally in lap-shear specimens of Alclad 7075-T6 aluminium alloy with different thicknesses, i.e., 0.8 mm and 1.6 mm. The joints were tested under low-cycle and high-cycle fatigue tests. Detailed observations of the fatigue fracture characteristics were conducted using a scanning electron microscope (SEM) with energy dispersive X-ray spectroscopy (EDS). The locations of fatigue failure across the weld, fatigue crack initiation, and propagation behaviour are discussed on the basis of the SEM analysis. The possibility of predicting the propagation of fatigue cracks in RFSSW joints is verified based on Paris's law. Two fatigue failure modes are observed at different load levels, including shear fracture mode transverse crack growth at high stress-loading conditions and at low load levels, and destruction of the lower sheet due to stretching as a result of low stress-loading conditions. The analysis of SEM micrographs revealed that the presence of aluminium oxides aggravates the inhomogeneity of the material in the weld nugget around its periphery and is a source of crack nucleation. The results of the fatigue crack growth rate predicted by Paris's law were in good agreement with the experimental results.

**Keywords:** aircraft industry; aluminium alloy; friction stir spot welding; mechanical engineering; single-lap joints

## 1. Introduction

In recent years, Refill Friction Stir Spot Welding (RFSSW) technology [1], which is a derivative of Friction Stir Spot Welding (FSSW), is one the main solid-state joining techniques, which are being extensively studied in the literature. RFSSW is seen as an attractive technique for joining difficult-to-weld aluminium alloys in aircraft structures [2–5]. One of the crucial assessment qualities in the aircraft industry is the fatigue performance of structural components and, therefore, many efforts have been made to investigate the fatigue properties of various grades of friction stir welded joints of aluminium alloys. Fatigue loading is the main cause of failure of weld joints and is evaluated either by fatigue crack propagation behaviour [6,7] or stress vs. the number of cycles to failure (S–N)

behaviour [8,9]. The results of the investigations of many authors revealed that the fatigue strength of friction stir welded joints is superior to fusion welded joints [10,11] and is equal to or less than base metal [12,13].

Various results on the influence of welding conditions on the fatigue strength of friction spot welded joints are reported in the literature. James et al. [14] reported that lower welding speed exhibits higher fatigue strength. Cavaliere et al. [9] reported an increase in fatigue strength with increase in welding speed. In contrast, Ericsson and Sandstrom [15] reported that fatigue properties are relatively independent of welding speed.

Most of the previous studies investigated the fatigue properties of joints between similar aluminium alloys fabricated by friction stir welding. Sivaraj et al. [16] evaluated the fatigue crack growth of 7075-T651 aluminium alloy joints. They found that the fatigue life of joints is appreciably lower than that of the unwelded base metal, which is attributed to the dissolution of precipitates in the weld zone. The effect of base metal temper conditions on the fatigue behaviour of friction stir welded joints of AA7039 aluminium alloy was studied by Sharma et al. [17]. It has been found that better tensile properties and superior fatigue strength are observed in AA7039 aluminium alloy joints in W-temper conditions. Effertz et al. [18] studied the fatigue life of FSSW 7075-T76 aluminium alloy using the Weibull distribution. They found that, for a relatively low load, corresponding to 10% of the maximum supported by the joint, the number of cycles surpasses $1 \times 10^6$. Hence, an infinite life can be attributed to the service component.

Many of the previous studies have focused on determining the influence of welding parameters on the failure modes of joints under quasi-static loading conditions and the weld zone microstructure. Hassanifard et al. [19] introduced a new method for enhancing the life and strength of FSSW joints using a localised plasticity process. The S-N data obtained revealed that the cold expansion method could improve the fatigue life of FSSW joints by up to six times. The microstructures and failure modes of friction spot welds in lap-shear specimens of aluminium 6111-T4 sheets were studied by Lin et al. [20]. The micrographs show that, under quasi-static loading conditions, the failure mainly starts from the necking of the upper sheet outside the weld. Under cyclic loading conditions, the micrographs revealed two types of fatigue cracks: growths into the lower sheet outside the stir zone and initiation of the crack from the bend surface of the upper sheet outside the weld. Finding a gap in the literature, Singh et al. [21] investigated the effect of the environment and welding speed on the fatigue properties of the FSSWed Al-Mg-Cr alloy. The study showed that the presence of oxides in the weld nugget zone in the case of liquid nitrogen cooling causes a significant increase in the fatigue crack growth rate, while the fatigue crack growth rate is lower in the case of welding. The observations of fatigue cracks of 6061-T6 aluminium lap-shear specimens under cyclic loading conditions suggest that the fatigue crack is initiated near the possible original notch tip in the stir zone and propagates along the circumference of the nugget. Then it propagates through the sheet thickness and finally grows in the width direction to cause a final fracture [22]. Moreover, the results indicate that the fatigue life predictions based on Paris's law agree well with the experimental results. Wang et al. [23] proposed improving the mechanical properties of 2024 aluminium alloy joints by using graphene nanosheets to strengthen the tip of the hook defect. According to the authors, the improvement of fatigue life originates from the combined effect of crack deflection, which results from its tight incorporation between the aluminium matrix and the graphene nanosheets. Kubit et al. [24,25] analysed the effect of structural defects on the fatigue strength of RFSSW joints using C-scan scanning acoustic microscopy and SEM. The observations of the fatigue fractures reveal that the clad layer at the bottom of the weld is a source of initiation of fatigue cracking. Insufficient plasticisation of sheet material is a crucial defect that influences the number of destructive cycles. It was found that the fracture mechanism depends on the value of load amplitude. Shahani and Farrahi [26] investigated the mechanical behaviour of the 6061-T6 aluminium alloy FSSW joints. It was concluded that the stress intensity factors and kink angle change during the growth of the fatigue crack. Therefore, the contact kink angle assumed in the literature is not an appropriate assumption. Investigations presented by Chowdhury et al. [27]

were aimed at evaluating the fatigue properties of FSSWed 5754-O aluminium alloys. At higher cyclic loads, nugget pull-out failure occurred since the fatigue crack propagated circumferentially around the nugget, while, at lower cyclic loads, fatigue failure occurred perpendicularly to the loading direction. Failure modes of spot friction welds in cross-tension specimens of 6061-T6 aluminium alloy sheets have been investigated by Lin et al. [28]. Paris's law for fatigue crack propagation has been successfully used for accurate development of the fatigue crack growth model. An overview of the fatigue mechanism, fatigue life assessment, crack growth rate, and factors influencing the fatigue of friction stir welded aluminium alloy joints has been provided by Li et al. [29].

The RFSSW solid-state technique eliminates the disadvantages usually observed in other spot-like joining technologies, such as the difficulty of automation, weight penalty, requirement to drill holes (riveted and bolted joints), and the presence of a keyhole originated by the FSSW [18,30]. The advantages of the RFSSW technique such as no additional material being required to fabricate the joint mean that it is ideal for joining structural components made of lightweight alloys in the aircraft industry. Lightweight aluminium alloys are characterised by high fatigue strength [31,32]. However, the welding process causes non-uniformity in the weld structure, which produces a stir zone (SZ), a thermo-mechanically affected zone (TMAZ), and a heat affected zone (HAZ). In recent work [24,25], the authors have shown that the structure of the joint in the Alclad sheet is characterised by the presence of a bonded ligament causing discontinuity in the weld structure. The presence of weld defects affects the fatigue failure mechanism. In the case of linear friction stir welding (LFSW), many scientific papers have been written on the fatigue strength of LFSW joints [33–35]. However, the issue of the fatigue cracking mechanism for RFSSW joints is still not adequately described, and only a few scientific studies deal with this issue [35,36].

In this paper, the RFSSW technique is used to join Alclad 7075-T6 aluminium alloy plates of various thicknesses, i.e., 1.6 mm and 0.8 mm. The single-lap joints were subjected to a fatigue test. Typical failure mechanisms for low stress-loading conditions and high-stress loading conditions are presented and discussed. The location of fatigue crack initiation and the propagation behaviour are also discussed based on the scanning electron microscopy (SEM) analysis. Furthermore, one of the main aims of the work was to verify the possibility of predicting the propagation of fatigue cracks in RFSSW joints using Paris's law.

## 2. Experimental Methodology

### 2.1. Material

Alclad 7075-T6 aluminium alloy sheets were selected as the test material. Sheets of two thicknesses corresponding to the thickness of the stringers (1.6 mm) and skin plate (0.8 mm) of the fuselage of an aircraft were joined using RFSSW. The main alloying elements of the 7075-T6 aluminium alloy are zinc, magnesium, and copper. While the alloy is characterised by high static and fatigue strength, it is difficult to cold form because the value of its yield stress is close to its tensile strength [7]. In addition, 7075-T6 aluminium alloy is difficult to weld and is characterised by a relatively low corrosion resistance [5,7]. However, the basic feature of this alloy is its combination of high mechanical strength and low weight, which makes it a material commonly used in the manufacturing of aircraft structures. Due to its relatively high susceptibility to intergranular corrosion, sheets used in the construction of thin-walled aircraft structures are usually clad, i.e., covered with a thin layer of pure aluminium (approximately 5% of the thickness of the base metal) in the hot rolling process.

The table lists the basic mechanical properties of the sheets tested, which were determined in the uniaxial tensile test using a Z100 universal testing machine (Zwick/Roell, Ulm, Germany). The following parameters were determined through the tensile test: yield stress $R_{p0.2}$ = 413.7 MPa, ultimate tensile stress $R_m$ = 482.6 MPa, and elongation at failure $A$ = 7%. Three specimens were tested and the average values of each specific parameter were determined. The chemical composition of 7075-T6 aluminium alloy is listed in Table 1.

**Table 1.** The chemical composition (in weight percentage) of 7075-T6 material.

| Al | Zn | Cr | Cu | Mg | Mn | Fe | Si | Ti | Other | |
|---|---|---|---|---|---|---|---|---|---|---|
| | | | | | | | | | Each | Total |
| 87.1–91.4 | 5.1–6.1 | 0.18–0.28 | 1.2–2.0 | 2.1–2.9 | max. 0.3 | max. 0.5 | max. 0.4 | max. 0.2 | 0.05 | 0.15 |

### 2.2. Welding Process

The fatigue tests were conducted on the single-lap RFSSW joints presented in Figure 1. The welding was conducted using an RPS100 spot welder by Harms & Wende GmbH & Co KG (Hamburg, Germany). Before welding, the workpieces were cleaned with acetone on the faying surfaces. The RFSSW process consists of the following stages.

- The clamping ring is fixed on the top surface of the upper sheet, and the tool stays there for a certain amount of time to produce the initial frictional pre-heating (Figure 2a),
- the sleeve plunges the sheet to the desired depth, and, at the same time, the pin moves in the opposite direction (Figure 2b),
- after reaching the desired plunge depth, the directions of movement of both the sleeve and pin begin to reverse (Figure 2c),
- the weld cycle is completed by removing the tool from the surfaces of the sheets (Figure 2d).

The values of the welding parameters are crucial in assuring the fabrication of welds without defects such as voids in the weld corner, bonding ligament, and structural discontinuities. The RFSSW parameters were selected based on the authors' experience and the results of previous investigations of the authors presented in the paper [37]. The first stage of the poly-optimisation of the values of welding process parameters was focused on the analysis of the tool plunge depth $g$ on the load capacity of the joint. For this purpose, the welds were fabricated at a tool plunge depth equal to 1.3, 1.5, 1.7, and 1.9 mm, with three different tool rotational speeds $n$ = 2000, 2400, and 2800 rpm. The load capacity of the joints obtained ranged from 5510 N to 8000 N. In the next part of the research, the focus was put on determining the influence of tool rotational speed and the duration of welding on the load capacity of joints. For the durations of welding $t$ = 1.5 s and 2.5 s, an increase of tool rotational speed causes an increase in the load capacity of the joint. For the duration $t$ = 1.5 s, an increase in the tool rotational speed from 2000 to 2800 rpm results in an increase in load capacity by 17.9% (from 6870 to 8100 N), while, for the welding time $t$ = 2.5 s, it does this by 11.5%. During the investigations, the hypothesis that the factor analysed did not have an influence on the load capacity of the connection that was tested. For the analysis, a confidence interval of 95% and $\alpha$ = 5% (the probability of rejecting the hypothesis tested, assuming it is true) was used. The dependent variable was the maximum shear load, while the process parameters were set as the independent variables. The values of the parameters that were selected assured the most favourable joint quality in terms of the highest static strength and quality of the weld structure (minimum number of structural discontinuities, structural notches, as small as the heat affected zone, etc.) [37].

The values of the welding parameters are as follows: tool rotational speed $n$ = 2400 rpm, duration of welding $t$ = 2.5 s, and tool plunge depth $g$ = 1.55 mm.

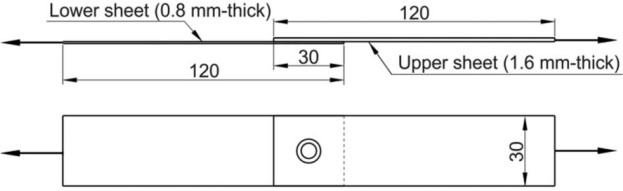

**Figure 1.** Geometry and dimensions of the specimen for the tensile test.

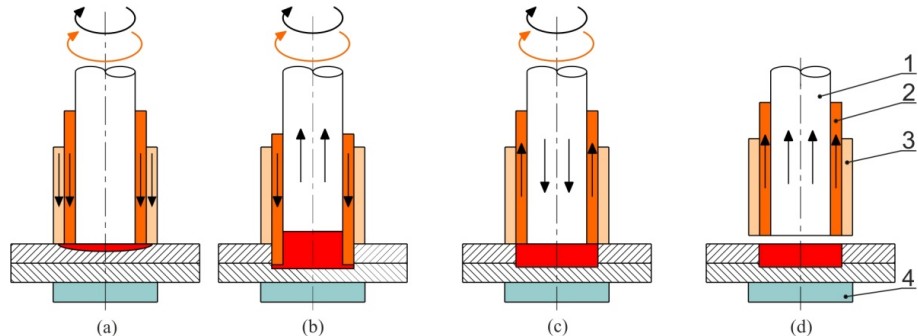

**Figure 2.** Stages of the Refill Friction Stir Pot Welding (RFSSW) process: touchdown and preheating (**a**), plunging (**b**), refilling (**c**), and retreating (**d**): 1-pin, 2-sleeve, 3-clamping ring, and 4-bracking anvil.

### 2.3. Fatigue Test

Fatigue strength tests were carried out on an Instron ElectroPulsTM E10000 testing machine (Instron, Norwood, MA, USA) designed for dynamic and static testing on a wide range of components and materials. Fatigue tests were carried out at room temperature with the following conditions: a frequency of 50 Hz and a limited number of cycles equal to $2 \times 10^6$. The coefficient of the stress cycle of $R = 0.1$ used is often used in aircraft component testing [38] and corresponds to a tension-tension cycle in which $\sigma_{min} = 0.1\sigma_{max}$. Four specimens were tested for each level of amplitude.

### 2.4. Characterisation of Fracture Surfaces

Microstructure and fracture morphologies of selected specimens were analysed using a S-3400N (Hitachi, Chiyoda, Japan) variable pressure scanning electron microscope (SEM). The samples were etched using Keller's solution (2 mL HF, 3 mL HCL, 5 mL $HNO_3$, and 190 mL $H_2O$). When analysing fatigue fractures, an analysis of the chemical composition was performed on selected areas of the fracture surface to verify fracture areas and identify impurities in the weld. The chemical composition analysis was conducted using energy dispersive spectrometry (EDS). Energy-dispersive X-ray (EDX) spectra were collected on the Quanta 3D 200i (FEI Company, Hillsboro, OR, USA) scanning electron microscope.

## 3. Results and Discussion

### 3.1. Microstructure Analysis

The cross-section of the RFSSW joint (Figure 3) can be divided into four regions in terms of the microstructural characteristics of the joint in sequence from the stir zone (SZ) toward the base material (BM), the thermo-mechanically affected zone (TMAZ), and the heat-affected zone (HAZ). In the SZ, a homogenous fine-grained microstructure was found, which was characterised by fully dynamically recrystallised equiaxial grains. In the HAZ, the grains are similar to those of the BM, but, in the direction of the TMAZ, the grains become slightly coarse. Within the centre of the SZ in the RFSSW process, a fine-grained microstructure is observed. In the case of the TMAZ, a high gradient of a grain size change is revealed. On the boundary of these grains, ultrafine precipitates were observed. Literature data suggest that these are likely particles of $Al_7Cu_2Fe$, Mg $(Zn, Cu, Al)_2$, and also $MgZn_2$ strengthening phases that occur in the 7xxx series aluminium alloys [25].

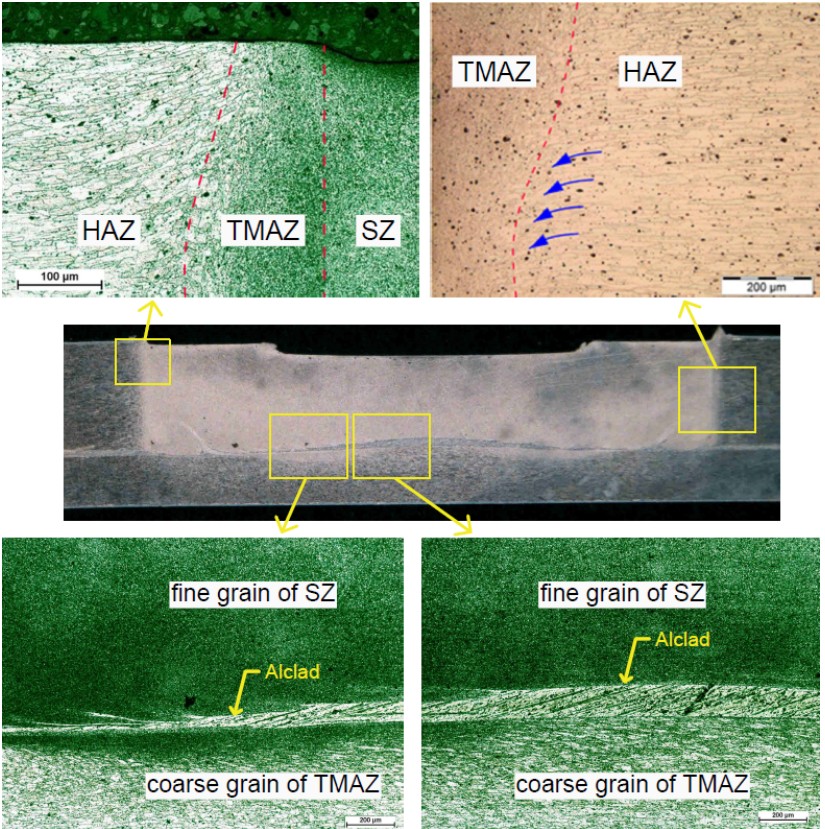

**Figure 3.** Cross-sectional view of the weld made with the parameters *n* = 2400 rpm, *t* = 2.5 s, and *g* = 1.55 mm.

The weld made using RFSSW consists of two main parts: the nugget of the joint and the HAZ. The nugget of the weld has a fine grain structure. This indicates the occurrence of the recrystallization caused by both high heat and pressure. In the bottom part of the weld, a plated coating (Alclad) is observed. Alclad within the weld nugget, which covered the sheet metals to protect them against corrosion, causes heterogeneity in the structure of the weld. These heterogeneities reduce the weld strength and clearly indicate the direction of flow of the material. The Alclad has a higher thermal conductivity (229 W/mK) where BM (134 W/mK) transfers the heat out of the joint very quickly. The Alclad layer also exists in the lower zone of the joint which, despite reaching a much higher temperature, is not degraded.

*3.2. Fatigue Diagram*

A fatigue diagram with areas corresponding to the specific failure mechanism is shown in Figure 4. Based on the results of shear fatigue testing of the joints, two basic mechanisms of fatigue failure were observed. A limited number of cycles to failure of $10^5$ is adopted to be the boundary between low-stress and high-stress fatigue loading. In the area of high-stress fatigue strength, the joints were mainly damaged by shearing in the plane in which the sheets were joined. High-stress fatigue strength was characterised by relatively high plastic deformation of the joint. The weld structure is weakened by a clad layer called a bonding ligament, which is composed of pure aluminium compressed at the bottom of the weld (Figure 5a). The bonding ligament originating from the Alclad at the original lap interface has been deformed and redistributed due to plasticised metal flow during the welding process. The bonding ligament, especially near the axis of the weld, is an inherent problem of the RFSSW joints [24,37].

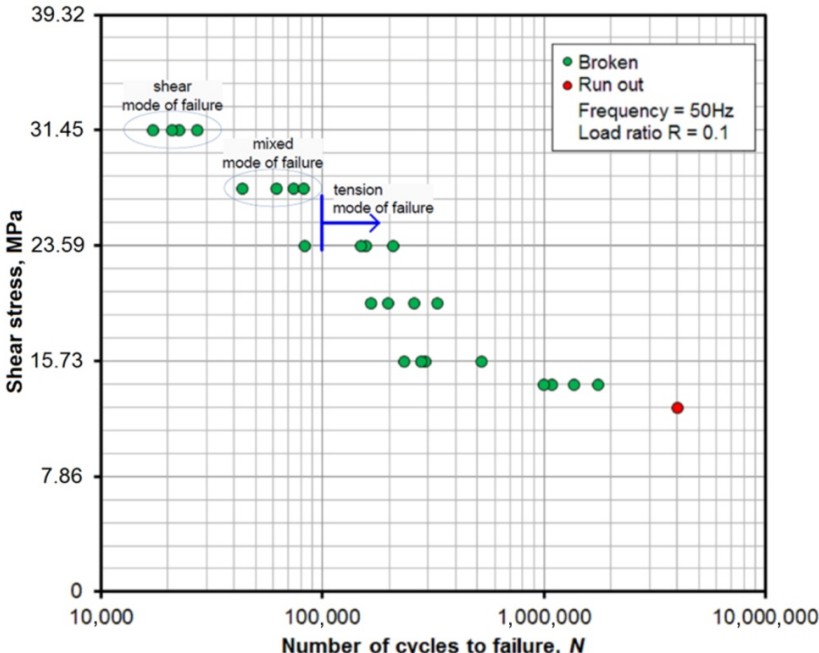

**Figure 4.** Fatigue diagram for Refill Friction Stir Spot Welding (RFSSW) joints subjected to high-cycle fatigue strength with a limited number of cycles equal to $4 \times 10^6$.

### 3.3. Fatigue with a High Value of Stress

Plastic deformations of the weld resulting from fatigue cycles are the cause of stress concentration at the boundary of the clad material in the weld structure and the fine-grained base metal (BM) in the SZ zone. Moreover, a thin and brittle layer of oxides was formed in the welding process as a result of thermo-mechanical changes in the BM. The presence of aluminium oxides aggravates the inhomogeneity of the weld structure around its periphery and is a source of crack nucleation. The oxides are considered to contaminate the weld. Aluminium and its alloys are very reactive with atmospheric oxygen and form a layer of oxide on its surface [39]. The oxide layer can cause entrapped oxide flaws in the stir zone. The existence of oxides in the nugget zone is confirmed by EDS analysis of the fracture surface of the joint in the area of the upper sheet that underwent fatigue shearing after 22,348 cycles with a maximum variable force value of 2,000 N. Figure 5b shows a fragment of the fracture surface with three characteristic areas of BM, a clad layer, and the boundary surface between the clad and BM containing a significant amount of oxides. The chemical composition in these areas is revealed by EDS spectra: EDS_1 (Figure 5c), EDS_2 (Figure 5d), and EDS_3 (Figure 5e), respectively.

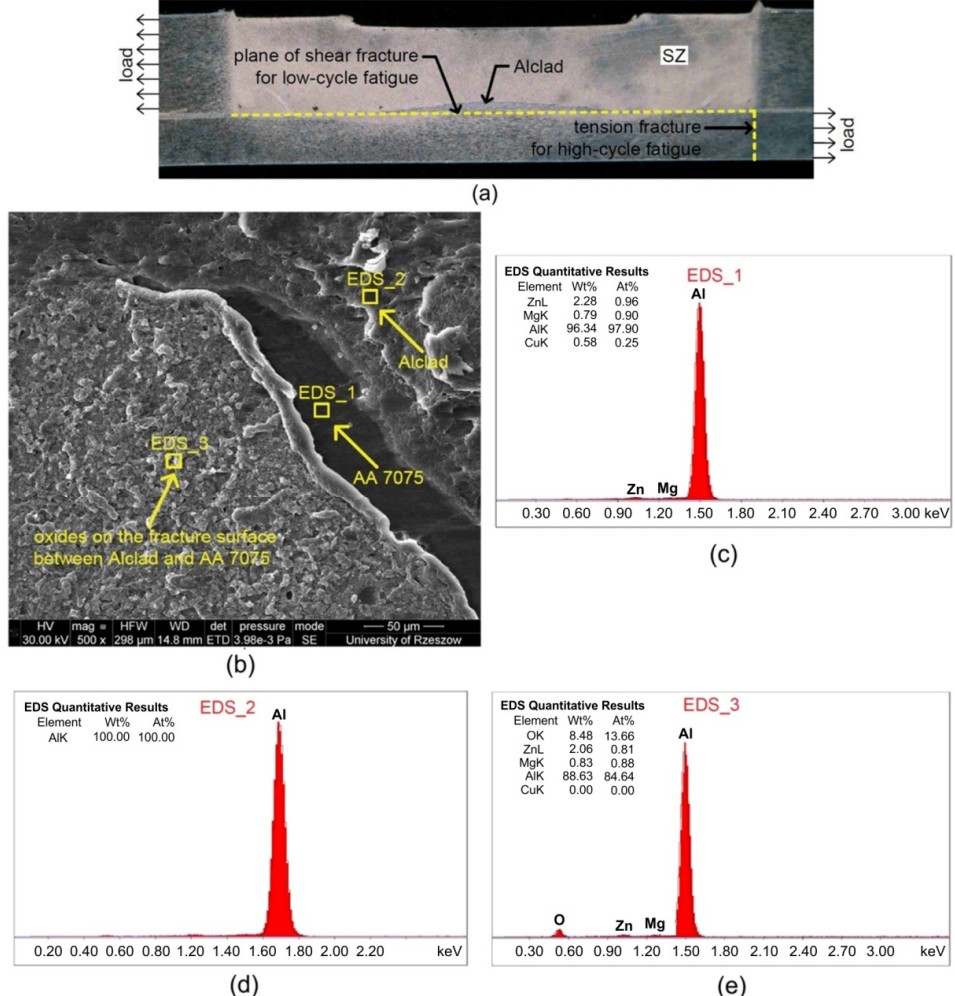

**Figure 5.** (**a**) Cross-sectional view of RFSSW joint structure, (**b**) scanning electron microscope (SEM) micrograph of fatigue fracture for a joint that was destroyed after 22,348 cycles at a stress level of 31.45 MPa and energy dispersive X-ray spectroscopy (EDS) spectra of weld material in areas EDS_1 (**c**), EDS_2 (**d**), and EDS_3 (**e**) in Figure 4b.

For a variable force with a maximum value of 2000 N, all the test samples exhibited the shear model of failure. Figure 6a shows the fracture surface in the shear plane of the lower sheet for a joint that was destroyed after loading more than 22,348 cycles. The presence of oxides on the surface of the clad material in the nugget zone was demonstrated on the basis of EDS spectra of fatigue fractures (Figure 6b–d). A high content of oxides on the fracture surface was also observed in the case of other samples that were destroyed as a result of shearing. Figure 7 presents a SEM micrograph of a sample that was destroyed after loading more than 26,913 cycles.

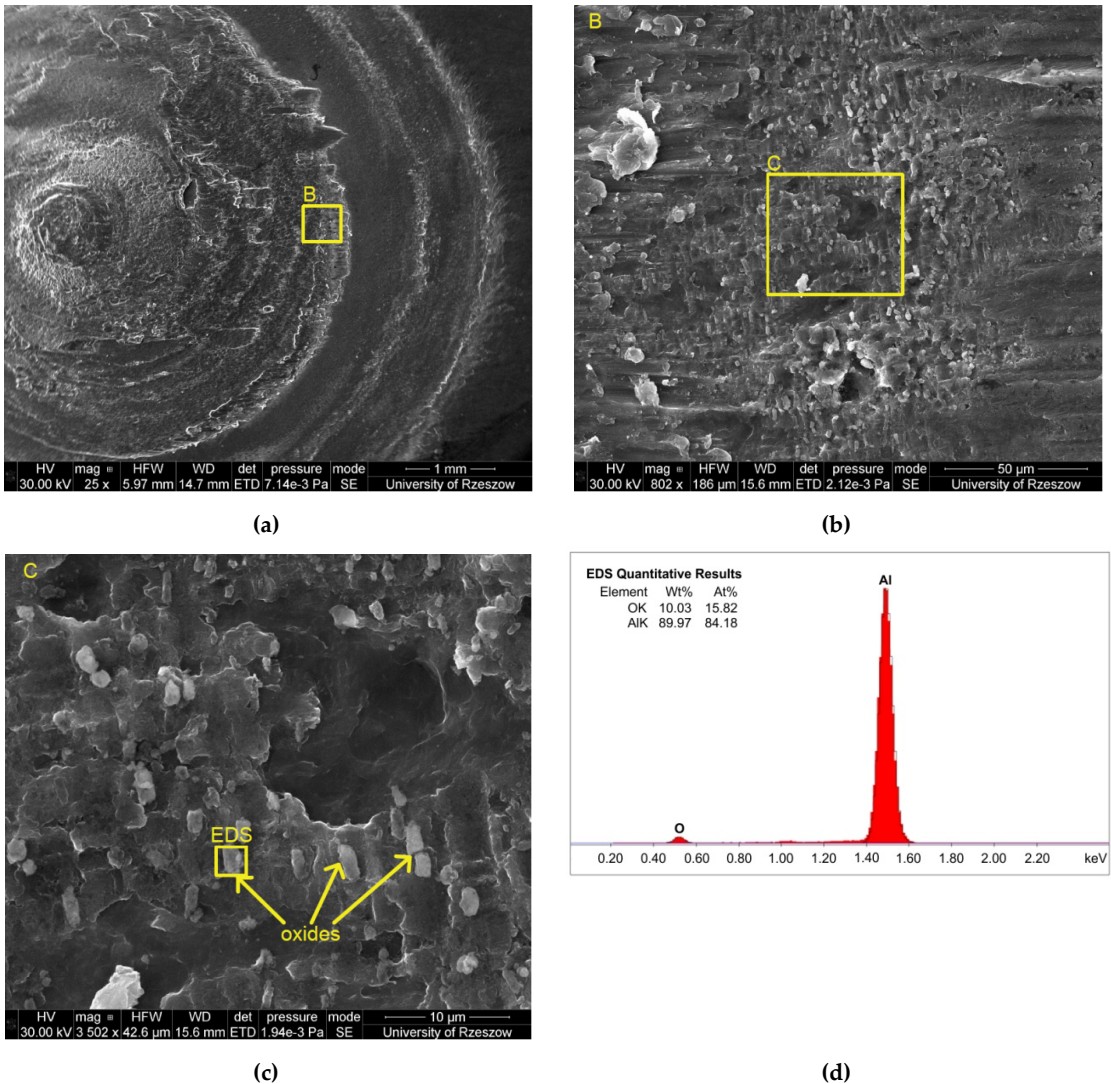

**Figure 6.** (**a**) Scanning electron microscope (SEM) micrograph of a fracture surface for a joint that was destroyed after 22,348 cycles. At a stress level of 31.45 MPa (**b**) and (**c**) magnified views in areas A and B, (**d**) EDS spectra of the aluminium oxides shown in Figure 5c.

In the case of selected specimens, especially those characterised by higher fatigue life from among those loaded with a variable force with a maximum value of 2000 N, SEM analysis has shown that, despite the fact that the samples were destroyed by shearing, the second mechanism of fatigue cracking progressed in parallel to the shear fracture. In the lower sheet, transverse cracks were disclosed, which were the result of stretching this sheet. This type of crack was usually initiated at the boundary of the SZ and TMAZ in the vicinity of the axis of the joint, and propagated toward the edges of the sheets. In the first phase of fatigue loading, the fatigue crack propagated along the weld perimeter, after which it took a linear direction. Figure 8a shows the fatigue fracture for a joint, which led to damage after 22,348 load cycles.

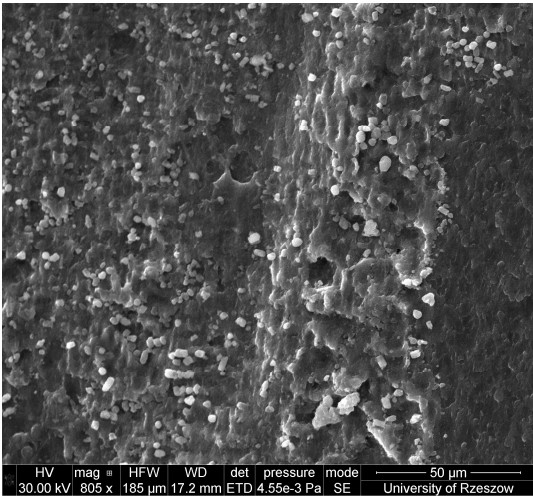

**Figure 7.** SEM micrograph of a fracture surface for a joint that was destroyed after 26,913 load cycles at a stress level of 31.45 MPa.

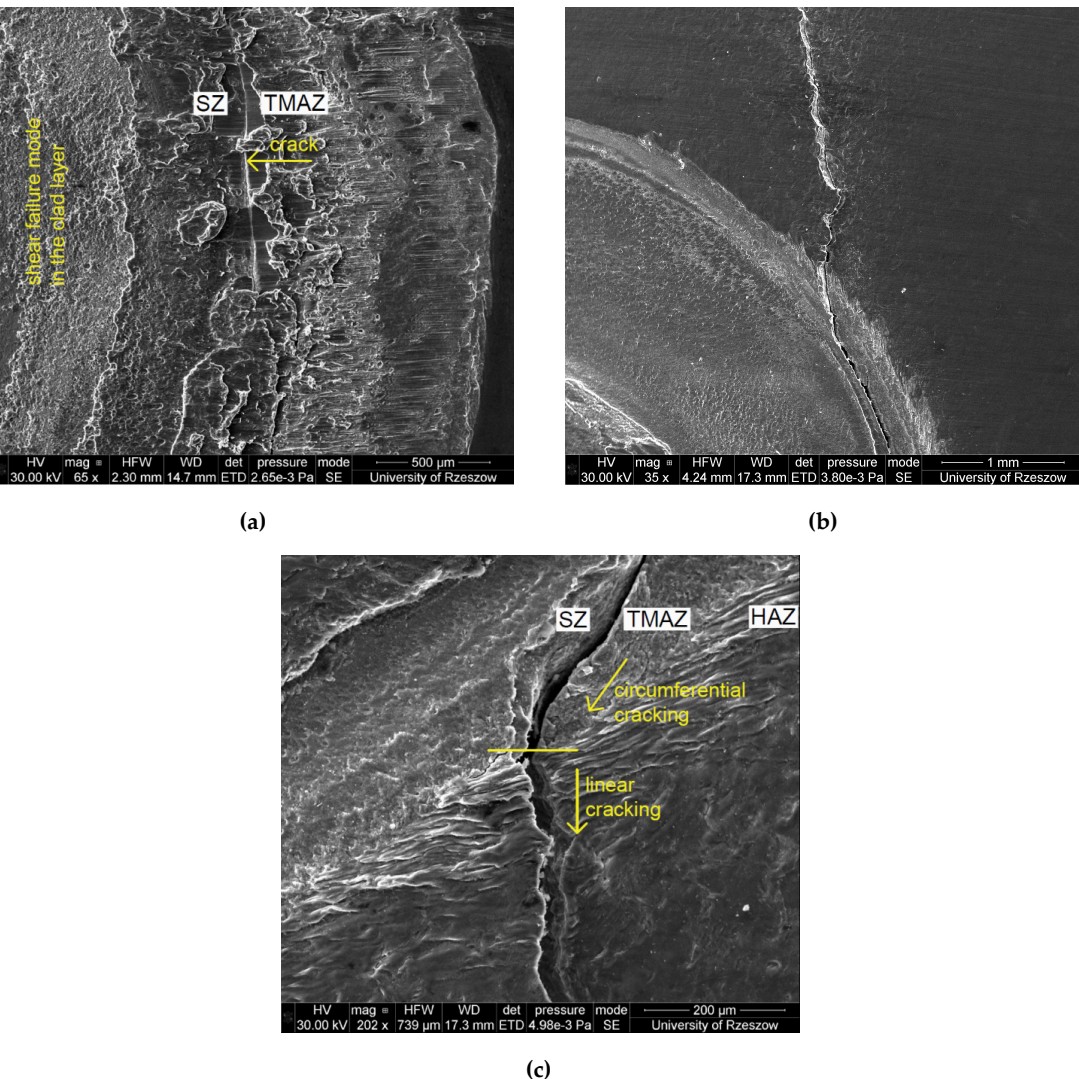

**Figure 8.** SEM micrographs of a fracture surface for joints that were destroyed after (**a**) 22,348 cycles at a stress level of 31.45 MPa, (**b**) 26,913 cycles at a stress level of 31.45 MPa, and (**c**) 42,639 cycles at a stress level of 27.52 MPa with a clear transverse fatigue crack in the lower sheet.

The joint was destroyed as a result of shearing at the boundary of the clad material and through the clad material. However, the crack of the lower plate was simultaneously initiated along the weld perimeter. Figure 8b shows the propagation of the crack as a result of tensile stress. The change in the direction of crack propagation from peripheral to transverse is clearly visible. A similar phenomenon was observed for a joint that had been destroyed as a result of shearing more than 42,639 cycles with a variable force with a maximum value of 1,750 N (Figure 8c). The results obtained are consistent with the study of Venukumar et al. [40] who tested the fatigue behaviour of friction stir spot welding refilled by the friction forming process (FSSW-FFP) lap-shear specimens, and concluded that, under high-stress loading conditions with lower load ranges, more transverse cracks were observed when compared to those with higher load ranges.

### 3.4. Fatigue with a Low Value of Stress

In the low stress range of fatigue strength, determined by the number of $10^5$ cycles with variable force, all test specimens were destroyed as a result of stretching the lower sheet (Figure 9), which was thinner than the upper sheet. In the low stress range of loading conditions, there are no plastic deformations caused by subsequent load cycles with a variable load. Hence, there is no crack initiation at the boundary between the clad material lying at the bottom of the weld and the fine-grained material.

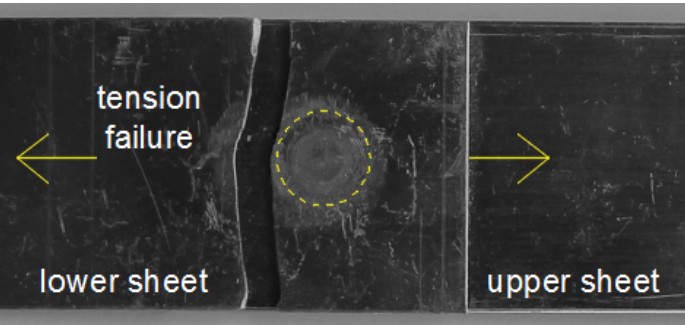

**Figure 9.** A typical pattern of destruction of the weld under high-cycle loading conditions.

### 3.5. Fatigue Crack Growth Rate

Due to the long-term operation of the RFSSW structure, knowledge about their high-cycle fatigue life is very important. An accurate description of fracture mechanisms due to fatigue during long-term operation, and, thus, the prediction of crack propagation, is an important issue. Therefore, it was decided to attempt to predict the speed of crack propagation in RFSSW joints based on well-established hypotheses. The authors propose the use of Paris's law in order to describe the propagation of fatigue cracks in the lap-shear specimens considered in this article. Paris's law for crack propagation has been successfully adopted to predict the fatigue crack growth rate of FSSW aluminium alloy joints [22,41]. In this paper, it was assumed that the initiation of the fatigue crack occurred at the boundary of the SZ and TMAZ zones. In addition, the initial crack propagation in the peripheral direction of the weld is not described by Paris's law.

The SZ, TMAZ, and HAZ have different properties in relation to the native material. In the RFSSW process, the material in the previously mentioned zones were subjected to thermo-mechanical change. This caused grain refining and modification of the mechanical properties of the material as a result of high temperature. Therefore, the mechanisms accompanying fatigue cracking and its initiation in the area are omitted in this analysis. The analysis was started when the direction of crack propagation was already linear and was propagating toward the edge of the sheets. Paris's law was valid for the linear cracking in the quasi-transverse direction to the load direction. Figure 10 shows in schematic fashion the area of validity of Paris's law (or Paris-Erdogan law) for the lap-shear specimens analysed. In this graph, the rate of crack growth *da/dN* is expressed on a logarithmic scale, and the amplitude of

the stress intensity factor $\Delta K$ is expressed on a linear scale. The $\Delta K$ is a material constant at a given stress ratio $R$. The length of the initial crack was adopted in the analysis as "pre-crack" (Figure 11). The threshold region in the propagation in Figure 10 does not occur for the $\Delta K$-values below the threshold level. The Paris region corresponds to stable macroscopic crack growth. The final failure region is associated with the high crack growth rate prior to failure.

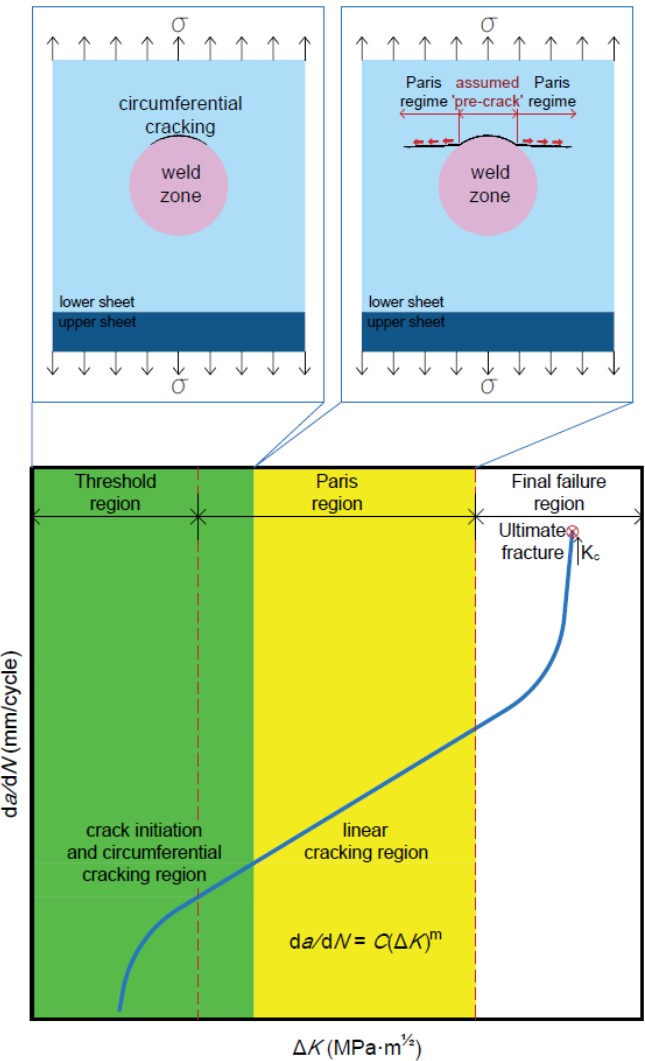

**Figure 10.** Rate of fatigue-crack growth with the assumed range of validity of Paris Law for lap-shear RFSSW specimens.

Paris and Erdogan [42] proposed that the relationship between the rate of fatigue-crack growth and the stress-intensity range is as follows.

$$\frac{da}{dN} = C \cdot (\Delta K)^m \tag{1}$$

$$\Delta K = \Delta\sigma \cdot \sqrt{\pi \cdot a} \tag{2}$$

$$\Delta\sigma = \sigma_{max} - \sigma_{min} \tag{3}$$

$$N = \int_{a_1}^{a_2} \frac{1}{C \cdot \left(\Delta\sigma \cdot \sqrt{\pi \cdot a}\right)^m} da \tag{4}$$

where *da/dN* is the fatigue crack growth rate, *N* is the life or number of cycles, *a* is crack length, $\Delta K$ is the stress intensity factor range, and *C* and *m* are constants proposed to incorporate the effects of material, mean load, loading frequency, and environment.

The fatigue crack growth rate is defined as the increase in its length during one loading cycle, and the case depends on the gap length, working stress, and material properties. Changes in the loading stress $\sigma$ during the load cycle cause corresponding changes in the stress in the area close to the edge of the crack, and, thus, changes in the stress intensity factor range $\Delta K$ (Equation (2)). Changes in the stress near the crack edge determine its behaviour.

The detailed procedure for obtaining feasible and comparable constants in Paris's law is defined in standards BS 6835:1988 and ASTM 647-95a. The values of constants *C* and *m* for 7075-T6 aluminium alloy used in this research are as follows: $C = 7 \times 10^{-11}$, $m = 4$.

Experimental verification was carried out thanks to which it was possible to confirm the accuracy of Paris's law for the selected fatigue area of the RFSSW joint. For selected specimens subjected to fatigue loading, measurements of the distances between the ends of cracks (Figure 11) were conducted during the tests. These measurements were carried out when the direction of crack propagation in the lower sheet changes from peripheral to quasi-transverse. These distances were assumed as 'pre-cracks' and then based on Paris's law. The theoretical courses of crack propagation were determined according to Equations (1)–(4).

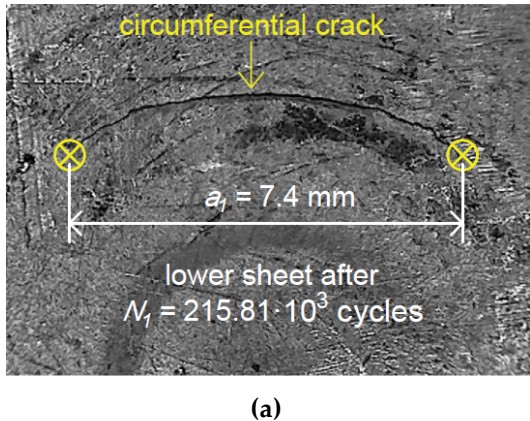
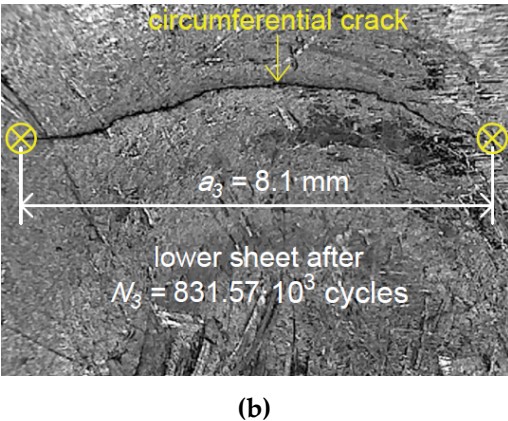

(a)                                                                (b)

**Figure 11.** Views of fatigue cracks observed during the test when the crack changes its direction of propagation from peripheral to quasi-transverse for joints that were destroyed after (**a**) 215,810 cycles and (**b**) 831,570 cycles

The predicted evolution of crack length according to Paris's law is shown in Figure 12. In general, the experimental results are well predicted using the Paris equation for specific loading conditions and geometrical configurations. This conclusion is in accordance with the results of Theos [43] who conducted fatigue analysis of welded aircraft wing panels. The highest consistency was obtained for the joint, which cracked at the number of cycles of $N = 289.82 \times 10^3$. The weld was resistant to destruction when the Paris law underestimates the number of cycles to failure *N*. The Paris equation has some limitations, i.e., asymptotic behaviour in both the threshold and final failure region is not described with this equation. Moreover, the stress ratio is not accounted for the crack growth. A significant portion of the fatigue life is occupied in the subcritical crack growth region described by Paris's law, particularly structures constructed from sheet or plates.

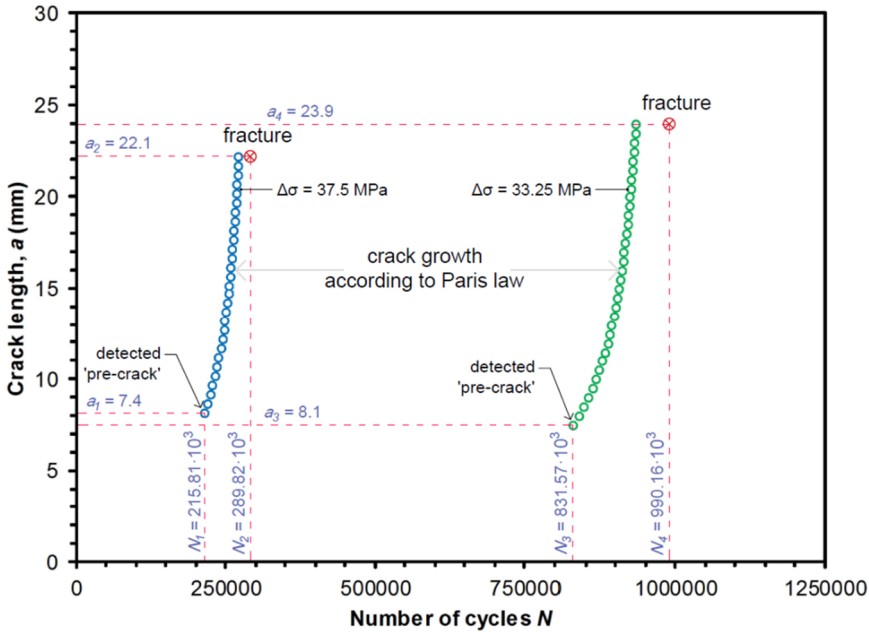

**Figure 12.** Fatigue crack growth behaviour with indication of the area of validity of Paris law for a lap-shear RFSSW joint loaded at a shear stress level of 15.73 MPa (blue points) and of 14.15 MPa (green points).

Analysis of SEM micrographs of fatigue fractures (Figure 13) shows that, in the area of the lower sheet, the propagation of the fatigue crack was approximately uniform throughout its thickness. This may prove that the assumption of Paris's law in the area of linear cracking of the joint is correct. Good consistency of the experimental results with the predicted number of cycles at which the joint would be destroyed allows one to conclude that the assumptions are correct. The considerations presented may form the basis for broader analyses, which predict crack propagation in RFSSW structures.

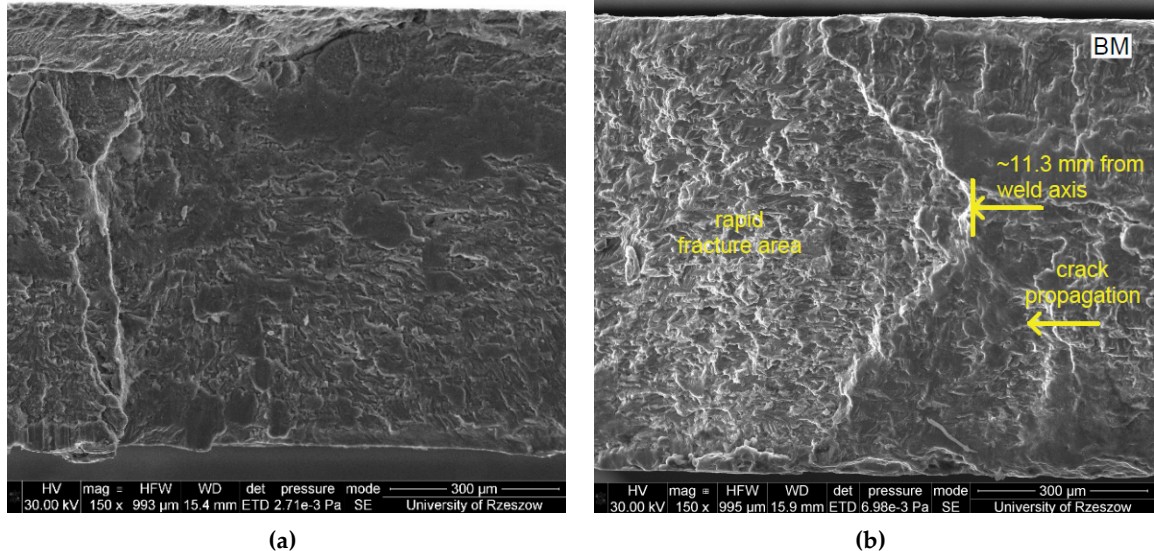

**Figure 13.** Fatigue fractures of lower sheets of joints that were destroyed as a result of stretching after (**a**) 231,794 cycles at a stress level of 15.73 MPa and (**b**) 1,363,043 cycles at a stress level of 14.15 MPa.

## 4. Residual Stresses Induced by Friction Stir Welding

Residual stresses are defined as self-equilibrating forces that remain in a material once all external forces have been removed. They arise due to complex thermal-mechanical-metallurgical interactions during welding. Residual stresses (RS) are results of complex thermal-mechanical- metallurgical interactions during welding. It may result in distortion of the weldments, which can lead to fitting issues and/or failures. Friction stir welding (FSW) also leads to the introduction of RS in welded structures. According to the Williams and Steuwer [44], residual stresses can be classified in terms of length scales on which they act over.

- type-I: extend over macroscopic areas and are averaged over several grains,
- type-II: extend between grains or sub-regions of grains and averaged over these areas,
- type-III: act on the inter-atomic level such as around inclusions or dislocation.

Several approaches for evaluating FSW-induced residual stresses have been proposed in literature based on the experimental techniques and numerical approaches. Residual stresses induced by friction stir welding are considered to be a major driving force for crack propagation if friction stir welded the structures [45]. The dominant role of RS on crack growth rates in 2024-T351 aluminium alloy joints was explicitly pinpointed by Bussu and Irving [46]. Their analysis underlined that the secondary role is played by the local microhardness and microstructure. The trend of crack propagation behaviour in aluminium alloys based on residual stress measurement has been discussed by Hong et al. [47]. The role of residual stresses and microstructure on crack propagation in 6063-T5 aluminium alloy joints has been experimentally investigated by Tra et al. [48]. It was found that crack propagation is mainly driven by the inhomogeneous microstructure in and around the welded area, whereas the effect of RS is not significant. A completely opposite conclusion was pointed out by Pouget and Reynolds [49]. They concluded that accurate growth rate predictions can be achieved by including a residual stress effect into the calculation. The key role of process-induced residual stresses on crack growth in FSW joints has also been shown by Citarella et al. [50] and Sepe et al. [51]. Difficulties in experimentally obtaining all the components of residual stress tensor significantly reduce the potential application of experimental methods [46].

Recently, some attempts to numerically assess the fatigue behaviour in FSW structures were discussed in literature. A numerical investigation on the influence of residual stress, induced by the FSW process, on fatigue crack growth in 2024-T3 aluminium friction stir welded butt joints has been proposed by Carlone et al. [45]. They found that, if the initial crack starts from the weld line, the process-induced opening stresses have an accelerating effect on the crack propagation. The implemented finite element method-dual boundary element method (FEM-DBEM) proved to be able to effectively predict multiple crack growth in the presence of residual stresses induced by the fabrication process. The influence of FSW process parameters on the residual stresses distribution has been also considered in recent investigations by Carlone et al. [52]. Results of FEM-DBEM-based computations revealed that the FSW process induced compressive residual stresses that play a delaying effect on the crack propagation if the initial notch is far enough from the weld line. Zadeh et al. [53] analysed three dimensional simulations of fatigue crack growth in friction stir welded joints of 2024-T351 aluminium alloy. Results of computations verified with experiments and the analytical method show 90% accuracy in terms of fracture mechanics and fatigue life prediction.

The effective numerical approaches able to predict the residual stresses and crack growth in friction stir welded joints are highly desirable. Therefore, further research will address the need for experimental and numerical analysis of the residual stresses in RFSSW joints.

## 5. Conclusions

This paper presents the results of fatigue life assessment of RFSSW single-lap joints subjected to high-cycle fatigue tests. The following conclusions are drawn from the research.

- As far as fatigue strength in low cycle conditions is concerned, the joints were mainly damaged by shearing in the plane in which the sheets were joined.
- The bonding ligament is the main element of the RFSSW joint weakening the fatigue strength of the joint.
- The RFSSW joint of Alclad sheets is contaminated by a high content of aluminium oxides. The presence of aluminium oxides aggravates the heterogeneity of the material in the weld nugget around its periphery and is a source of crack nucleation. Oxides could partially break down and be released from the outside of the weld area during the welding process.
- As far as fatigue strength is concerned in low stress-loading conditions, determined by the number of $10^5$ cycles with a variable force, all test specimens were destroyed as a result of stretching the lower sheet.
- Paris's law for crack propagation has been successfully adopted to predict the fatigue crack growth of lap-shear RFSSW specimens. Although some assumptions have been made, the comparison of the analytical and experimental fatigue crack growth rate confirms the potential of Paris's law to analyse the crack growth in RFSSW joints.

**Author Contributions:** Conceptualization, A.K., M.D., and W.B. Investigation, A.K., M.D., and W.B. Formal analysis, A.K., M.D., and W.B. Validation, A.K., M.D., and W.B. Interpretation of the results, all authors contributed equally. Writing–original draft, A.K., and T.T. Writing–review & editing, T.T. Project administration, Ľ.K. and J.S. Funding acquisition, Ľ.K. and J.S. All authors have read and agreed to the published version of the manuscript.

**Funding:** The Slovak Research and Development Agency under grant number APVV-17-0381 – Increasing the efficiency of forming and joining parts of hybrid car bodies – funded this research. And the Grant Agency of the Ministry of Education, Science, Research, and Sport of the Slovak Republic grant number VEGA 1/0259/19.

**Acknowledgments:** The authors are grateful for the support of experimental works to the Slovak Research and Development Agency under project APVV-17-0381 – Increasing the efficiency of forming and joining parts of hybrid car bodies, and the Grant Agency of the Ministry of Education, Science, Research, and Sport of the Slovak Republic grant number VEGA 1/0259/19. The present work was carried out with the generous support of the Belgian Welding Institute (Zwijnaarde—Ghent, Belgium). The authors of this paper would like to kindly thank ir. Koen Faes for preparing the welded joints.

**Conflicts of Interest:** The authors declare no conflict of interest.

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
