# Peer review of "Fatigue Life Assessment of Refill Friction Stir Spot Welded Alclad 7075-T6 Aluminium Alloy Joints"

_metals, doi:10.3390/met10050633_

Round 1

Reviewer 1 Report

The submitted manuscript deals with an experimental investigation on the fatigue crack propagation in lap joints manufactured by refill friction stir spot welding of aluminum alloys.

The analyzed topic is interesting and well included between the main scope of the journal. The manuscript is quite readable and clear, The reported results are valuable from an academic and industrial point of view.

Referee is, in general, positive toward this submission, nevertheless some drawbacks deserve further attention, as detailed in what follows:

  • the first part of the introduction is not adequately focused. Extraneous sentences sohuld be removed in order to make this section more compelling;
  • the whole analysis is based on joints welded an unique combination of parameters. From this point of view the paper appears quite weak. the choice of processing parameters sohuld be properly motivated. Authors are als oinvited to extend the analysis to different combinations of parameters;
  • the analysis of the fatigue behavior is noe exhaustive since the influence of residual stresses on crack propagation rate was completely neglected. Even if the welding induced stresses in solid state welding processes are lower if compared to fusion welding process, their influence on crack growth has been welll demonstrated. Authors are invited to extend their overview of the state-of-the-art as well as their discussion in this sense, considering, for instance previous contributions by Citarella and coworkers.

Major revisions are recommended

Reviewer 2 Report

Dear Authors,

Your study presents the fatigue behavior of Refill Friction Stir Spot welded Alclad 7075-T6, which is really interesting for Industrial applications.

High cycle fatigue testing is indeed time consuming and somehow expensive. therefore, the presented results could be useful for the sectors of industry and they could take advantage of it. However, the goal of the scientific papers should be innovation, which is not achieved in your work.

Also an important point before welding of aluminium alloys is to clean the surface of plates (using Alcohol, … ). It is not mentioned in your work.

The strength of Aluminium 7075 is significantly depended on the precipitations and their locations in the Aluminium matrix. This part is not included in your work. (microstructural characterisation after Welding).

The effect of welding process on the static performance and hardness behaviour are the key points for further studies like fatigue life assessments.

Although the submitted article presents the valuable results, I should reject it.

Reviewer 3 Report

1) The paper gives experimental fatigue results for number of cycles between 10000 and 2000000. This is the usual SN curve for welded joints (e.g. see Eurocode 3). Then it is not correct to speak of low cycle fatigue. The distinction  between the different failure behavior should be based instead on higher and lower stress ranges (or, if you prefer, on number of cycles higher or lower than 100000). Please change the presentation accordingly.

2) The stress range for the lower number of cycles in fig. 11 seems to be wrong: it should be higher than the one for higher number of cycles, and it is not compatible with the data reported in fig. 3.

Reviewer 4 Report

This paper explores the fatigue behaviour of some RFSSW joints applied to sheets of 7075 aluminium alloys. Both low cycle and high cycle fatigue behaviours are evaluated. The damage of the joints in different cycle modes are investigated using SEM and EDX. The main issue seems to come from the presence of aluminium oxides that contaminated the joints and decreases their performance in fatigue. A model is applied successfully to predict the fatigue crack growth.

The problem of joining of aluminium alloys is a very relevant topic especially in the aerospace industry. The paper is written in very good English and is scientifically sound. I recommend for publishing with no further corrections.

Round 2

Reviewer 1 Report

The previously submitted paper has been revised by the authors as per referees' comments.

The acceptance for publication of the paper is now recommended.

Reviewer 2 Report

Dear Authors,

Thank you so much for your revision.

Although your paper presents valuable results for industrial application, there is no scientific innovation included in your work. Presenting only high cycle fatigue performance of welded joint can not be enough for a scientific work. There is a paramount importance to add the microstructural characterization in order to improve the quality of your study.
